Optimization of fermentation conditions for physcion production of Aspergillus chevalieri BYST01 by response surface methodology

Zhang Shuxiang 1
Jiang Zhou 1
An Suwen 1
Jiang Xiaolan 2 jiangxiaolan@ahau.edu.cn
Zhang Yinglao 1 zhangyl@ahau.edu.cn
1 School of Life Sciences, Anhui Agricultural University , Hefei , China
2 State Key Laboratory of Tea Plant Biology and Utilization, Anhui Agricultural University , Hefei , China
Uversky Vladimir
Electronic publication date: 2024 Oct 30
Publication date: 2024
Volume: 12
Electronic Location ID: e18380
Received 2024 May 20; Accepted 2024 Oct 1
Copyright: © 2024 Zhang et al.
Copyright year: 2024
Copyright holder: Zhang et al.
License: This is an open access article distributed under the terms of the Creative Commons Attribution License, which permits unrestricted use, distribution, reproduction and adaptation in any medium and for any purpose provided that it is properly attributed. For attribution, the original author(s), title, publication source (PeerJ) and either DOI or URL of the article must be cited.
License URL: https://creativecommons.org/licenses/by/4.0/

Keywords: Aspergillus chevalieri, Physcion, Yield, Response surface methodology

Funding: National Natural Science Foundation of China (NSFC) 32102272 and 32270015 Open Fund of State Key Laboratory of Tea Plant Biology and Utilization SKLTOF20210106 This work was supported by the National Natural Science Foundation of China (NSFC) (32102272 and 32270015) and The Open Fund of State Key Laboratory of Tea Plant Biology and Utilization (SKLTOF20210106). The funders had no role in study design, data collection and analysis, decision to publish, or preparation of the manuscript.

==============================
This study aimed to optimize the culture conditions of the termite-derived fungus Aspergillus chevalieri BYST01 for the production of physcion, a characteristic component of the traditional herb rhubarb, which has been commercially approved as a botanical fungicide in China. First, potato dextrose broth was screened as the suitable basal medium for further optimization, with an initial yield of 28.0 mg/L. Then, the suitable carbon source, fermentation time, temperature, pH value, and the rotary shaker speed for physcion production were determined using the one-variable-at-a-time method. Based on the results of single factors experiments, the variables with statistically significant effects on physcion production were further confirmed using the Plackett-Burman design (PBD). Among the five variables, temperature, initial pH, and rotary shaker speed were identified as significant factors (P < 0.05) for physcion productivity in the PDB and were further analyzed by response surface methodology (RSM). Finally, we found that the maximum physcion production (82.0 mg/L) was achieved under the following optimized conditions:initial pH 6.6, rotary shaker speed of 177 rpm, temperature of 28 °C, and glucose concentration of 30 g/L in PDB medium after 11 d of fermentation. The yield of physcion under the optimized culture conditions was approximately threefold higher than that obtained using the basal culture medium. Furthermore, the optimum fermentation conditions in the 5-L bioreactor achieved a maximal physcion yield of 85.2 mg/L within 8 d of fermentation. Hence, response surface methodology proved to be a powerful tool for optimizing physcion production by A. chevalieri BYST01. This study may be helpful in promoting the application of physcion produced by A. chevalieri BYST01 to manage plant diseases.

Introduction

Phytopathogens pose a serious threat to global economy and environmental systems by causing crop yield loss and toxins pollution (Singh et al., 2023). For over a century, agricultural chemicals have been used for plant disease management and have resulted in dramatic yield benefits. However, the excessive use of agrochemicals poses a threat to the environment and human health because of the toxic residues that accumulate in the soil and enter the food supply. Thus, alternative methods to control phytopathogens are urgently required to ensure crop production and reduce economic losses to human societies (Tang et al., 2021). Biopesticides, which are typically based on the use of antagonistic microorganisms or their products, are promising ecofriendly alternatives for managing fungal and bacterial plant diseases (Elnahal et al., 2022).

One of the challenges of biological control is the reliability of its efficacy. The utilization of antagonistic microorganisms is often considered less dependable and efficient than chemical control, due to exposure to uncontrollable external environmental variables (Collinge et al., 2022). By contrast, metabolites derived from microorganisms or plant extracts are dependable and efficient for controlling the plant diseases. Screening for novel biocontrol agents applicable to plant disease management is a promising research area, and many metabolites with favorable biocontrol activities have been identified in recent decades (Raymaekers et al., 2020). However, only a few promising natural metabolites with convincing efficacy have been commercially developed. For example, validamycin, abamectin, kasugamycin and polyoxin are the main bioactive metabolites produced industrially by microorganisms and are widely applied as agro-antibiotics (Li et al., 2020). Low content is the main factor limiting the utilization of these active ingredients to control plant diseases. Improving the yield of bioactive natural products plays an important role in promoting their applications (Pham et al., 2019; Devi et al., 2023).

Single-factor methods and response surface methodology (RSM) are effective in improving production of microbial metabolites by optimizing suitable conditions. For example, compared with the initial culture medium, Zhang’s group significantly improved the production of avermectin B1a from 3.5 to 5.1 g/L after optimizing the medium components using RSM (Gao et al., 2009).

Physcion has been registered as a novel botanical fungicide in China with excellent plant disease prevention properties. However, some disadvantages severely hinder its application, such as low contents, long plant growth cycles, and costly purification procedures. Obtaining physcion from microorganisms can reduce costs compared with those of plant-derived physcion (Yao et al., 2023). In our previously study, we found that an insect-associated fungus, BYST01, could produce physcion with the highest yield, compared with that of the natural strains (Zhang et al., 2024). Here, for the first time, we efficiently improved the physcion yield of BYST01 through medium optimization using RSM. This foundational research promotes the acquisition of fungicidal physcion from microorganisms rather than from plants.

Materials and Methods

Strains

Aspergillus chevalieri BYST01 was early isolated from the gut of termite and deposited at China Center for Type Culture Collection with the number CCTCC2021285 (Zhang et al., 2024).

Culture media

According to the methods detailed previously (Suwannarach et al., 2019; Zhao et al., 2017), a piece of fungal mycelial from the slant was inoculated on PDA plate at 28 °C for 1 week, and then the periphery of the growing colony was cut into homogeneous plugs with a diameter of 5 mm. Five pieces of fresh plugs were added to flasks containing different liquid media to select the basal medium for the optimization of physcion production. The cultures were then fermented at 28 °C in a shaker rotating at 180 rpm for 11 days.

Nine liquid media, including potato dextrose broth (PDB, 200 g/L fresh potato and 20 g/L glucose, pH 7.0), malt-extract broth (ME, 20 g/L malt extract, 20 g/L glucose and 1 g/L peptone, pH 7.0), sabouraud dextrose broth (SDB, 10 g/L tryptone and 40 g/L glucose; pH 5.6), Oatmeal medium (OA, 30 g/L oatmeal), Czapek yeast extract broth (CZB, 3 g/L NaNO3, 1 g/L KH2PO4, 0.5 g/L MgSO4•7H2O, 0.5 g/L KCL, 0.01 g/L FeSO4 •7H2O and 30 g/L glucose, pH 7.0), Gause medium G-1 (GA1, 20 g/L soluble starch, 0.5 g/L NaCl, 1 g/L KNO3, 0.5 g/L K2HPO4·3H2O, 0.5 g/L MgSO4·7H2O and 0.01 g/L FeSO4·7H2O; pH 7.4), Takashio medium (Taka, 1 g/L KH2PO4, 1.0 g/L MgSO4·7H2O, 2 g/L glucose and 1 g/L tryptone, pH 7.0), yeast extract phosphate broth (YPB, 1 g/L yeast extract, 6.0 g/L KH2PO4, 4 g/L NaH2PO4 and 1 g/L NH4OH, pH 7.0) and dextrose peptone medium (DPM, 60 g glucose, 20 g/L peptone, 2 g/L KH2PO4, and 1 g/L MgSO4•7H2O, pH 7.0) were used for screening the basal medium.

Quantification of physcion by HPLC-DAD analyses

For shake flask fermentation, the fungus A. chevalieri BYST01 was cultivated on PDA plate at 28 °C for 7 days and about five plugs with diameter of 5 mm were transferred into the nine media described above each 250 mL Erlenmeyer flask containing 100 mL liquid medium. After cultivating at 28 °C and 180 rpm for 11 days, the yield of physcion in different medium was observed by HPLC analyses.

After fermentation, 2 mL of chromatographic methanol was added to equal volume of fermentation broth, and then the mixture was centrifuged for 10 min. Further filter the supernatant with 0.22 μm membrane filter. The filtrate is used for subsequent HPLC analysis. Chromatographic analysis was performed by using a C18 reverse-phase analytical column: Fisher Wharton Xbridge C18, 5 μm, 10 mm * 250 mm. SHIMADZU LC16 parameters were as follows: the column oven temperature was set at 40 °C, the injection volume was 5 μL with a flow rate of 1.0 mL/min. The mobile phase consisted of deionized water and 100% methanol at a ratio of 15% to 85% (v/v) for 20 min. For quantification, standard physcion (0.50 mg) was dissolved in 1.0 ml of DMSO and used as a standard stock solution for generating calibration curves. The stock solution was diluted 1/2, 1/4, 1/8, 1/16, 1/32 and 1/64 in DMSO solution to afford gradient solutions. The calibration curves were made from the physcion solutions using a linear fit for the relationship of area vs. concentration of physcion at UV 254 nm.

Single factor experiments for physcion production

The PDB medium was chosen as the basal medium for further optimization for physcion production under different medium compositions (carbon source, nitrogen source and inorganic salts, etc.). Carbon source, such as glucose, sucrose, maltose, xylose and fructose were tested to examine their effects on the physcion production. The glucose (20 g/L) in the normal PDB medium was replaced by these carbons at the same concentration. And the influence of different concentrations of the carbon (glucose, 10–40 g/L) were also analyzed. Different nitrogen source supplements and inorganic salts source were set to evaluate their effects on the physcion production. The extra nitrogen source, such as glycine, urea, NH4Cl, (NH4)2SO4 and yeast extract were individually added at 0.3% (w/v) concentration. The extra inorganic salts source (CH3COONa, MnCl2, NaCl, GaSO4, KCl and FeCl3) were individually added at 0.05% (w/v) concentration.

Moreover, to improve the physcion production, the fermentation conditions were optimized at different fermentation time (7, 9, 11, 13, 15 and 17 d), temperature (20, 22, 24, 26, 28, 30 and 32 °C), the initial pH (4.0, 4.5, 5.0, 5.5, 6.0, 6.5, 7.0, 7.5 and 8.0) and the speed of the rotary shaker (90, 120, 150, 180, 210 and 240 rpm).

The fermentations were performed in 250 mL of medium containing of 100 mL medium broth. The experiments were performed in triplicate, and the mean values were used to minimize variations.

Optimization by Plackett-Burman design (PBD)

The Plackett-Burman design (PBD) is a normal method used to identify the important factors in the early stage of response surface methodology (RSM) (Wu et al., 2022; Zaki et al., 2019). An initial screen was carried out for above five single factors (different fermentation time, glucose concentration, temperature, pH and the speed of the rotary shaker) using a PBD experimental design (Design-Expert 10.0.3). The design set was shown in Table 1 and statistical data was shown in Table 2. Each independent variable is represented in two levels, high and low, which are denoted by (1) and (−1), respectively. Twelve runs of various levels of factors were formulated by the software and the output response represented the yield of physcion. The t test was used to determine the significance of the regression coefficients. The variable with a P value less than 0.05 was considered as a significant factor (Tajik et al., 2024).

Table 1 Matrix of the Plackett-Burman design and the corresponding experimental data of physcion production values for the Aspergillus chevalieri BYST01 strain.

Run	A-temperature (°C)	B-rotating speed (rpm)	C-glucose concentration (g/L)	D-pH	E-fermentation time (d)	Actual response physcion yield
(mg/L)	Predicted physcion yield (mg/L)	
1	30	210	20	7	13	18.14	18.70	
2	26	210	40	6	13	5.21	5.64	
3	30	150	40	7	7	10.65	11.07	
4	26	210	20	7	13	11.61	11.47	
5	26	150	40	6	13	2.65	1.15	
6	26	150	20	7	7	6.24	6.41	
7	30	150	20	6	13	8.48	10.94	
8	30	210	20	6	7	18.14	14.85	
9	30	210	40	6	7	10.63	12.29	
10	26	210	40	7	7	7.53	8.33	
11	30	150	40	7	13	13.44	11.65	
12	26	150	20	6	7	2.88	3.14	

Table 2 Statistical data for the determination of variable significance in the Plackett-Burman design experiment.

Components	Sum of squares	Mean squares	Regression coefficient	F value	P value	Significance	
A-temperature (°C)	156.67	1	156.67	35.46	0.001	**	
B-rotating speed (rpm)	60.39	1	60.39	13.67	0.0101	*	
C-glucose concentration (g/L)	19.71	1	19.71	4.46	0.0791		
D-pH	32.08	1	32.08	7.26	0.0358	*	
E-fermentation time (d)	1	1	1	0.23	0.6515		
Model	269.85	5	53.97	12.22	0.0042	**	
Model R2 = 0.9160							
Model adjusted R2 = 0.8360							
Notes:

** P < 0.01.

* 0.01 < P < 0.05.

Optimization by response surface methodology (RSM)

Following the three variables (temperature, initial pH and the speed of a rotary shaker) that are capable of significantly influencing the physcion production were screened by the PBD, a useful response surface methodology (the Box-Behnken experimental design, BBD) was applied using the Design-Expert software to optimize the physcion production conditions. The three independent variables were analyzed at three levels (−1, 0, and 1) with 17 experimental runs in Table 3. Based on BBD, the optimized objective was to achieve the maximum physcion production using this software. Based on BBD, experiments with different fermentation conditions (temperature, initial pH and speed of a rotary shake) under a constant culture medium were conducted. The physcion yield was subjected to analysis of variance (ANOVA), and the experimental data of BBD were fit with the following second-order polynomial equation: Y = β0 − β1A − β2B + β3C + β12AB − β13AC − β23BC − β11A2 − β22B2 − β33C2. Where Y is the predicted physcion yield, A (temperature), B (speed of a rotary shake) and C (initial pH), are the initial independent variables; β1, β2 and β3 are the linear coefficients; β12, β13 and β23 are the interaction coefficients; and β11, β22 and β33 are the quadratic coefficients.

Table 3 Experimental design and results of the Box–Behnken design for optimization of the physcion production by BYST01.

Run	A-temperature
(°C)	B-rotating speed
(rpm)	D-pH	Experimental physcion yield (mg/L)	Predicted physcion yield (mg/L)	
1	26	150	6.5	32.06	37.32	
2	30	150	6.5	6.9	9.32	
3	26	210	6.5	22.61	20.19	
4	30	210	6.5	6.78	1.52	
5	26	180	6	20.24	19.33	
6	30	180	6	11.58	13.52	
7	26	180	7	49.18	47.25	
8	30	180	7	5.48	6.39	
9	28	150	6	13.76	9.4	
10	28	210	6	8.57	11.9	
11	28	150	7	38.09	34.76	
12	28	210	7	2.97	7.33	
13	28	180	6.5	79.18	79.04	
14	28	180	6.5	82.25	79.04	
15	28	180	6.5	80.71	79.04	
16	28	180	6.5	80.48	79.04	
17	28	180	6.5	72.56	79.04	

Based on the results of the suggested optimal culture conditions by the soft, a new experiment was performed under these optimal parameters. The target physcion yield obtained using these conditions present in Table 3 was compared with that obtained using the second-order polynomial equation calculated.

Bioreactor fermentation

According to the shaking flask experiment results, the optimized fermentation medium was further verified in 5.0 L stirred fermenter (BIOTECH-5BG; Shanghai Baoxing Bio-Engineering Equipment Co., Ltd., Shanghai, China) filled with 2.0 L broth. The strain plugs were cultured in PDB for 3 days prepare seed culture. Then 20% seed culture was transferred to the optimized media. The fermentation was incubated at 28 °C with a stirring speed of 180 rpm and pH 6.6. The physcion yields obtained in fermenter and flask were then compared.

Results and discussion

The basal medium selected for optimization

Total of nine media were used to select the basal medium for optimization. After a 11 d fermentation progress, HPLC quantification of production showed that PDB was the optimal medium for production of physcion with a yield of 28.0 mg/L. The yield of physcion in ME (18.4 mg/mL) and SDB (15.8 mg/mL), which were lower than that of PDB (Fig. 1A). Only trace physcion was detected in GA1 and Takashio medium, while no physcion was detected in YPB and OA media (Fig. 1A). The biomass of the strain cultured in different media were evaluated and showed no significant correlation with production of physcion (Fig. 1B). Thus, PDB was designated as the basal medium for following fermentation conditions optimization.

Figure 1 Effects of medium composition on physcion and biomass production of Aspergillus chevalieri BYST01.

(Left) Physcion. (Right) Biomass.

The effect of single factor experiments of physcion production

Common parameters (such as carbon source, nitrogen source, inorganic salts, temperature, initial pH, fermentation time and the speed of the rotary shaker) have a certain effect on physcion production, and these different parameters were tested by one single factor method.

As showed in Fig. 2A, a significant difference of the physcion yield was observed by the PDB consisting with different carbon source at the same concentration of 20 g/L. As the glucose was used as the carbon source added to the PDB, the maximum physcion production (30.0 mg/L) was presented. In addition, the glucose concentration of the PDB had an important role in producing physcion. Results in Fig. 2B indicated that the physcion yield reached its highest productivity (43.0 mg/L) at glucose concentration of 30 g/L. Figure 2C indicated that extra nitrogen source adding to PDB had a negative impact on production of physcion. The physcion productivity of BYST01 were decreased under these adding extra nitrogen source (glycine, urea, NH4Cl, (NH4)2SO4 and yeast extract) with 3.5–12.7 mg/L, which were much lower than that of normal PDB (28.0 mg/L). In addition, the tested inorganic salts (CH3COONa, MnCl2, NaCl, GaSO4, KCl and FeCl3) also inhibited the physcion production (Fig. 2D). The fermentation time had a certain influence on the accumulation of physcion. Early termination of fermentation process will affect the physcion production, while long-term fermentation will cause waste of resources and even lead to the metabolism and decomposition of physcion, thus reducing the production. As presented in Fig. 2E, the yield of physcion reached a peak at 11 d with a value of 40.0 mg/L in normal PDB consisting with 30 g/L glucose, and then decreased.

Figure 2 Effects of nutrient and fermentation factors on physcion production by Aspergillus chevalieri BYST01.

(A) Carbon source. (B) Glucose concentration. (C) Nitrogen source. (D) Inorganic salts source. (E) pH. (F) Fermentation time. (G) Fermentation temperature. (H) Agitation speed of flasks.

For screening the optimal temperature for fermentation, a series of temperature was set. The production of physcion was sensitive to the temperature. It could be seen in Fig. 2F that with the increase of temperature, the yield of physcion increased gradually, and reached the maximum at 28 °C, with the yield of 40.0 mg/L. When the temperature was over 28 °C, the yield of physcion decreased. The production of physcion was evaluated at different pH. The results showed that with the gradual increase of pH, the yield of physcion also gradually increased. The maximum yield was reached at pH 6.5 with a value of 74.0 mg/L, and this pH was the optimal pH for production of physcion (Fig. 2G). Following the same procedure, the influence of agitation rate on the accumulation of physcion was observed. The highest physcion production was achieved at 180 rpm. (Fig. 2H).

Thus, the optimal fermentation parameters were fermented in PDB consisting with 30 g/L glucose at 28 °C and pH 6.5 with continuous shaking at 180 rpm for 11 days by single factor experiments.

Plackett-Burman design (PBD)

Based on the results of the single-factor experiments, the main factors, such as glucose concentration, temperature, initial pH, fermentation time, and the speed of the rotary shaker, that may play an important role in the production of physcion, were further analyzed using PBD experiments (Table 1).

The results of the PBD were statistically analyzed, as shown in Table 2. To determine the optimum response, a fitted first-order model for physcion production was obtained from the PBD as follows: Y (“Y” represents physcion yield, mg/L) = 9.63 + 3.61A + 2.24B − 1.28C + 1.64D + 0.29E, where Y is the predicted physcion yield; A, B, C, D, and E, are the coded factors of temperature (°C), rotating speed (rpm), glucose concentration (g/L), pH, and fermentation time (d), respectively. A factor with a confidence level greater than 95% (P < 0.05) was considered to have a significant effect on physcion yield and was selected for further study. The linear regression coefficient (R2) was 0.9160, the adjusted R2 was 0.8360, and the P-value was 0.0042 (P < 0.05) for the model. These results indicated that the model is suitable for the PBD experimental design. Based on the PBD results, among the five studied variables, temperature, initial pH, and the speed of the rotary shaker were the three factors significantly correlated with the yield of physcion (P < 0.05).

Response surface methodology (RSM)

Three combinations of components, temperature, initial pH, and rotating speed were selected for optimization using RSM, and the experimental physcion yields were recorded. The results of 17 runs from the Box-Behnken experiments for the effects of three independent variables are represented in Table 3. The maximum experimental value for physcion production was 82.25 mg/L based on the RSM. The predicted response (Y) for physcion production obtained from further linear multiple regression analysis of the experimental data was described as follows: Y = 79.04 − 11.67A − 6.23B + 5.2C + 2.33AB − 8.76AC − 7.48BC − 28.09A2 − 33.86B2 − 29.33C2, where Y is the predicated physcion yield, and A–C are the temperature, rotation speed, and initial pH, respectively. The statistical significance of this equation was confirmed using Fisher’s F-test. The model F-value of 61.52 indicated that the model was significant (Table 4). There was only a 0.01% chance that such a large F-value could occur due to noise. The linear regression coefficient R2 was 0.9875, and the adjusted R2 was 0.9715 for the model. These results indicated that the model was suitable for explaining the experimental parameters and their interactions. Of the first terms (A, B, and C), temperature (A), rotation speed (B), and initial pH (C) had significant effects on the physcion yield (P < 0.05), with the P-values of 0.0004, 0.0122 and 0.0267, respectively. Of the interaction terms (AB, AC, BC), temperature (A) interacting with initial pH (C) and rotating speed (B) interacting with initial pH (C) had significant effects on physcion yield (P < 0.05). The quadratic terms (A2, B2, and C2) had an extremely significant effect on physcion yield (P < 0.001).

Table 4 ANOVA for response surface quadratic regression model of physcion production.

Terms	Sum of squares	Degrees of freedom	Mean squares	F value	P value	Significance	
Model	15,311.68	9	1,701.3	61.52	<0.0001	**	
A-temperature (°C)	1,089.28	1	1,089.28	39.39	0.0004	**	
B-rotating speed (rpm)	311	1	311	11.25	0.0122	*	
C-pH	216.01	1	216.01	7.81	0.0267	*	
AB	21.76	1	21.76	0.79	0.4045		
AC	306.95	1	306.95	11.1	0.0126	*	
BC	223.95	1	223.95	8.1	0.0248	*	
A2	3,321.83	1	3,321.83	120.13	<0.0001	**	
B2	4,827.51	1	4,827.51	174.58	<0.0001	**	
C2	3,621.61	1	3,621.61	130.97	<0.0001	**	
Residual	193.57	7	27.65				
Lack- of- fit	136.39	3	45.46	3.18	0.1465		
Pure error	57.18	4	14.29				
Cor total	15,505.25	16					
R-squared	0.9875						
Adj R-squared							
Notes:

** P < 0.01.

* 0.01 < P < 0.05.

To better understand the effects of the variables and their interactions on physcion yield, three-dimensional and two-dimensional response surface curves were exhibited in Fig. 3, based on the constructed model. Three-dimensional graphs were generated for the combination of the two variables, keeping the other one at the optimum level determined by the path of the steepest ascent for physcion production by BYST01. The response surface was convex, suggesting that the optimum conditions were well defined, and there was a maximum for each variable. Response surfaces were generated as an interaction function of temperature, rotation speed, and pH and are presented as 3D surface responses in Figs. 3A–3C, respectively. The effects of temperature, rotation speed, and pH on physcion production are also presented in a 2D contour form (Figs. 3A–3C). Based on the RSM analysis, the predicted maximum production of physcion would be 81.12 mg/L, when the uncoded levels of the fermentation conditions were set at 27.5 °C with a rotary shacking speed of 176.5 rpm and an initial pH of 6.57. Validation of these predictions related to physcion production was confirmed by triplicate laboratory trials in flask culture, and the yield of physcion was 82.0 mg/L under the modified conditions (28 °C, 177 rpm, and pH 6.6). There was almost no difference from the predicted values.

Figure 3 Three-dimensional (left) and contour (right) graphs of different variables vs. physcion production of Aspergillus chevalieri BYST01.

(A, B) Three-dimensional and two-dimensional response surface plots showing the effect of fermentation temperature and agitation speed of flasks on physcion production. (C, D) Three-dimensional and two-dimensional response surface plots showing the effect of fermentation temperature and pH on physcion production. (E, F) Three-dimensional and two-dimensional response surface plots showing the effect of agitation speed of flasks and pH on physcion production.

Production of physcion at bioreactor scale

To evaluate the possibility of using the optimized fermentation medium for physcion production at a larger bioreactor scale, fermentation of BYST01 was carried out in a 5 L bioreactor. In this study, the optimized medium supported the maximum productivity of physcion at 85.2 mg/L on the eighth day of fermentation, which was consistent with that of the shaker flask. This indicated that the optimum culture conditions obtained by the RSM experimental design in submerged fermentation supported the enhanced production of physcion in a bioreactor.

Discussion

Physcion-producing microorganisms are promising sources due to their ability to utilize a simple medium, facilitate easy fermentation, and offer easy of use and sustainability, although traditional herb rhubarb-derived physcion has already been commercialized in China for controlling plant diseases (Ma et al., 2010; Qi et al., 2022). Various microorganisms have been reported to produce physcion in low quantities (Li et al., 2018; Sadorn et al., 2018; Parvatkar et al., 2009; Zin et al., 2017). Improving the yield of physcion produced by microorganisms is important for the application of beneficial strains. Various studies have reported that optimization of the key factors, such as fermentation mode, culture medium type and composition, fermentation temperature, time, pH, and rotation speed can activate the production of microbial metabolites (Manon Mani et al., 2023; Verma et al., 2023; Dhakshinamoorthy et al., 2021; Pu et al., 2013).

In the present study, PDB medium was screened as a suitable basal medium for optimization, and glucose concentration, fermentation temperature, time, rotary speed, and pH were considered positive factors affecting the accumulation of physcion for the first time using the one-factor-at-a-time optimization method. Xiong et al. (2020) reported that nitrogen and inorganic salts sources played an important role in regulating the yield of exopolysaccharides produced by microorganisms. Here, we evaluated the influence of both factors on the production of physcion and found that the nitrogen and inorganic salt sources tested in the PDB medium had negative effects on the accumulation of physcion produced by BYST01. This provides useful information for further studies on the regulatory mechanisms of physcion biosynthesis.

Statistical optimization of fermentation parameters is a comprehensive and useful tool for evaluating the effects of variouss factors on the fermentation process by planning experiments and building models (Jinendiran et al., 2019). RSM was an attractive strategy for optimizing the effect of these variable factors and has been successfully used for the optimization of fermentation process parameters, such as different carbon sources, nitrogen sources, inorganic salts, and pH for improving the yield of questin produced by A. flavipes HN4‑13 (Guo et al., 2020). Among the five positive variables, fermentation temperature, pH, and rotary speed were identified as the significant factors in the PBD experiments. The optimal levels of these three variables were further verified using the BBD of the RSM. The physcion yield increased to 82.0 mg/L under the optimized conditions selected after RSM, which was 2.9-fold higher than the initial yield. The results showed that RSM is a powerful tool for optimizing physcion production by A. chevalieri BYST01. To the best of our knowledge, this is the first report of enhanced physcion production by A. chevalieri BYST01 using RSM. This also suggests the potential of A. chevalieri BYST01 for the largescale production of physcion through fermentation.

In this study, PDB was selected as the optimal medium for physcion production by A. chevalieri BYST01. The increase in physcion production after optimization revealed that temperature, pH, and rotation speed significantly influenced the yield of active physcion in flask culture. The highest yield (82.0 mg/L) of physcion was achieved under the following optimized conditions: 100 mL culture broth in 250 mL Erlenmeyer flasks, consisting of 200 g/L potato, 30 g/L glucose, and the optimal fermentation conditions (initial pH 6.6, rotary shaker speed of 177 rpm, temperature of 28 °C for 11 d), which was 2.9-fold higher than the yield before fermentation optimization. The optimized medium was further verified in a 5 L stirred fermenter, and the production of physcion was 85.2 mg/L on the eighth day of fermentation. Therefore, physcion production was greatly enhanced by optimizing the fermentation strategy, which shows great potential for application in physcion production. This study may be helpful in promoting the application of A. chevalieri BYST01 in the management of plant diseases.

Supplemental Information

Supplemental Information 1 Media composition.

Supplemental Information 2 Culture conditions.

Additional Information and Declarations

Competing Interests

Author Contributions

Data Availability

The authors declare that they have no competing interests.

Shuxiang Zhang conceived and designed the experiments, performed the experiments, prepared figures and/or tables, and approved the final draft.

Zhou Jiang performed the experiments, prepared figures and/or tables, and approved the final draft.

Suwen An performed the experiments, prepared figures and/or tables, and approved the final draft.

Xiaolan Jiang analyzed the data, authored or reviewed drafts of the article, and approved the final draft.

Yinglao Zhang conceived and designed the experiments, prepared figures and/or tables, authored or reviewed drafts of the article, and approved the final draft.

The following information was supplied regarding data availability:

The raw data is available in the Supplemental Files.

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
