# Peer review of "Optimization of fermentation conditions for physcion production of Aspergillus chevalieri BYST01 by response surface methodology"

_PeerJ, doi:10.7717/peerj.18380_

## Round 0.1 · original submission · Major Revisions

· Academic Editor

Major Revisions

Please address all the critiques of both reviewers and amend manuscript accordingly.

·

Basic reporting

Overall, the writeup needs proper grammatical check.
The work is interesting and may be impactful. I recommend the following working to improve the manuscript.

1. Explain the novelty and research gap in this work as such type of work is already published many times. Add this at the end of the introduction section, Literature and references are sufficient . Critical spell check is required

Experimental design

2. How will you commercialize this work? Explain the complete holistic approach of commercialization including costing, feasibility plan. Also write complete SWOT analysis. Add all these details in the conclusion section. Further mention the reproducibility of your experimental data.

3. Add the high pixel TIFF images with better resolution. Figures captions should be concise and add the figures details in the main manuscript body.

4. Link this work with the SDGs especially climate change.
5. Add the latest citations of 2023 and 2024 and try to avoid older ones.

6. Write down the intelligent manufacturing technology/ automation involved in your product synthesis at bulk industrial scale.

Validity of the findings

Validity and findings have missing link.
7. Specific suggested changes are as follows:

i. Abstract: Mention RSM results as well.
ii. Line 55-56: “active in the improving the production” delete “the” before the word “production”.
iii. Line 63: Instead of using the sentence “.........when compared with that from plant” use “when compared with plant originated”.
iv. Line 64-66: give proof of the claim mentioned “However, to date, no natural microorganisms with high yield of physcion (>20 mg/L)”
v. Line 69, why termite isolated Aspergillus is used?
vi. Line 78-84: pH in several mentioned media like Takashio medium is missing.
vii. Line 86 what is meaning of the word used in heading “Quanlification”
viii. Line 90, how fermented medium was processed and sample was prepared for HPLC analysis, after fermentation of 11 days.
ix. Complete methodology must be mentioned.
x. Line 71-118: no reference is given for methodology
xi. Line 119-128: authors should mention the design they set, mention names of the factors and selected range. PBD is not clearly mentioned in the methodology section
xii. After applying PBD what is the need to apply BBD, ON what grounds three factors were selected in BBD design.
xiii. Line 129-146: under the heading of RSM, no details about factors, range, why this range selected etc must be given in Materials and methods section, which is missing.
xiv. Line 149: it must be cleared that whether 5L or 2L fermenter was used.
xv. Line 151: which two strategies are mentioned “two strategies were compared”
xvi. Please make clear why HPLC estimation was done. Is there any alternative cheaper estimation methods tested?
xvii. Line 159: do spell correction “mediumin”
xviii. Line 162: Use “on”, instead of “for” in the line The effect of single factor experiments for physcion production”.
xix. Line 195: spell check please “Plackette-Burman design”
xx. Line 202: what is the reading of Y in this line “Y (physcion yield, mg/L)”.
xxi. Line 208: The spelling of “Plackett-Burman” is different from line 200 where “Plackette-Burman” is used?
xxii. Line 447: Table 4, model interpretation must be given in text and in the table, it must be clear, that model was significant or not, and what about the Lack of fit. Nothing given in interpretation.
xxiii. Asterik sign “*” in the table is not decoded, that what does that mean, it must e clearly stated on the bottom of each table.
xxiv. IN Figure 2a, concentration of carbon sources must be mentioned
xxv. For bioreactor study, no data is presented in the figure/table, no results is mentioned, that how long the fermentation continues, what was the make and made of the fermenter, which medium was particularly used etc.

Additional comments

Scale up study of fermenter/bioreactor results are all missing except physcion units which is not enough to explain the experiments
IN the present for, it is needs major revision.

·

Basic reporting

english needs proofreading

Experimental design

acceptable

Validity of the findings

The author presented this article discussing the possibility of yield improvement of Physcion as a natural botanical fungicide. The topic may draw the attention of readers and could be accepted after addressing some major comments

1. The authors have not considered using solid state fermentation culture, which provides higher yield in these cases.
2. What other plant insecticides are available, why use physion only? Many plant extracts were reported to act as safe fungicides . see the following
Maj, W., Pertile, G., Różalska, S. et al. Comprehensive antifungal investigation of natural plant extracts against Neosartorya spp. (Aspergillus spp.) of agriculturally significant microbiological contaminants and shaping their metabolic profile. Sci Rep 14, 8399 (2024). https://doi.org/10.1038/s41598-024-58791-4


3. Spelling and grammar mistakes as in line 33 should be corrected and the whole manuscript should be revised by a native English speaker

---

## Round 0.2 · Major Revisions

· Academic Editor

Major Revisions

Please address issues pointed by the reviewer and revise manuscript accordingly.

·

Basic reporting

.

Experimental design

The experimental design is incomplete and highly ambiguous. Please see the comments below:
Line 74-76: The method for preparing the Aspergillus cubes is ambiguous: give following details:
1. Which sort of casts are used for preparing cubes?
2. How much is the concentration of Aspergillus in each cubes?
3. What standard method is used for checking the concentration of mycelium/spore/conidia etc in the cube.
4. How to check the health of the cube.
5. How Aspergillus is injected in the cube.
6. For how many days, it require to incubate the in-cube culture if done?
7. What was the storage temperature, humidity conditions for cube storage?
8. Have you checked the shelf life of the cube which is supposed to be used as seeds?
Line 79-89: Write complete sentence. “Nine media including……………………..”
Line 91: No explanation is given about PDA plate, how these are prepared, and why these are prepared and used instead of simple cell/conidial inoculum.
Line 91: If Shake flask fermentation is used, then where the Aspergillus plates were used?
Line 92: How “five plugs with 5mm diameter” are placed on the mouth of Erlenmeyer flask.
Line 94: there is no reference reported for the fermentation time period, how the 11 days period is used for the study?
Line 116: ‘the extra nitrogen source……”, there was no basic nitrogen source mentioned earlier in the experimental design, then how this is extra source?
Line 73-126: There is no reference for the methodology is mentioned at any place?
Line 128: replace “to identified” with “to identify”
Line 134-135: what is “by the soft and the response….”?
Line 155: where are these conditions mentioned “second order polynomial equation”?
Line 158: what does meaning of “strategy medium”?
Line 157-161: Under the heading “Bioreactor fermentation”, no reference is given, no model of bioreactor is mentioned, detailed experimental conditions were missing? No details about sample withdrawing is mentioned, all details are must. How much inoculum is used in the bioreactor.

Validity of the findings

Findings of the research paper are significant and presented well. please see following suggestions:

Line 168: what is meaning of word “mediumin”
Line 222: replace “Ph” with “pH”.
Line 331: make correction in reference, it seems two references are merged by mistake
Figure 3 is missing in the text, so its explanation is also missing

---

## Round 0.3 · Minor Revisions

· Academic Editor

Minor Revisions

Please address remaining concerns of the reviewer and amend manuscript accordingly.

·

Basic reporting

mentioned under experimental design heading

Experimental design

A good effort is made to address the comments, but still lot of ambiguity about authors basic experimental design regarding microbial experiments/fermentation.
1. It seems confusing to make plugs from a cylindrical hole punch inside the slant, as slants are made in tests tubes, it is not possible to make plugs of uniform size? Authors are advised to add pictures of making these uniform plugs or cubes?
2. It will question the reproducibility of the experiment if every time the cells are different, o researchers should check the conidia using Haemacytometer slide?
3. For research, it is not a satisfactory response.
8. line 91, What is PDB, it is not well known abbreviation?
8.line 94, For a pre experiment it recommended to do the experiment from exponential to decline phase of the microorganism. Pre-experiment can not be discontinued after getting highest production?
8. line 116, It is astonishing that researchers even don’t know that potato is not a nitrogen source, rather it is a carbohydrate source??THIS MAKES ALL THERESEARCH QUESTIONABLE?
8. LINE 134-135, Response not clear
8. line 157, the detailed fermenter conditions are mentioned, but still the reference is missing.

Validity of the findings

mentioned under experimental design heading

Additional comments

mentioned under experimental design heading

---

## Round 0.4 · accepted · Accept

· Academic Editor

Accept

All remaining issues were addressed, and the revised manuscript is acceptable now.